Evidence for sweep signatures in antibiotic-resistant strains in three species of bacteria

Pradhananga Anjani 1
Benitez-Rivera Lorena 1
Clark Candace 1
Tisthammer Kaho H. 2
Pennings Pleuni S. pennings@sfsu.edu 1
1 Biology, San Francisco State University , San Francisco , CA , United States of America
2 University of California, Davis , Davis , CA , United States of America
Thomas Jonathan
Electronic publication date: 2024 Oct 24
Publication date: 2024
Volume: 12
Electronic Location ID: e18178
Received 2024 May 21; Accepted 2024 Sep 4
Copyright: ©2024 Pradhananga et al.
Copyright year: 2024
Copyright holder: Pradhananga et al.
License: This is an open access article distributed under the terms of the Creative Commons Attribution License, which permits unrestricted use, distribution, reproduction and adaptation in any medium and for any purpose provided that it is properly attributed. For attribution, the original author(s), title, publication source (PeerJ) and either DOI or URL of the article must be cited.
License URL: https://creativecommons.org/licenses/by/4.0/

Keywords: Antibiotic resistance, Selective sweep, Soft sweep, Diversity, Escherichia coli, Staphylococcus aureus, Enterococcus faecium

Funding: NIH R01AI134195 NIH MARC T34-GM008574 NIH SFSU/UCSF MS Bridges to the Doctorate T32-GM142515 The Genentech Foundation #G-7874540 FR-015977 / P-13419 NIH MBRS-RISE R25-GM059298 Anjani Pradhananga and Pleuni S. Pennings were supported by NIH grant R01AI134195. Lorena Benitez-Rivera was supported by NIH MARC: T34-GM008574 and NIH SFSU/UCSF MS Bridges to the Doctorate: T32-GM142515. Lorena Benitez-Rivera and Anjani Pradhananga were supported by the Genentech Foundation: #G-7874540 and FR-015977 / P-13419. Candace Clark was supported by NIH MBRS-RISE: R25-GM059298. The funders had no role in study design, data collection and analysis, decision to publish, or preparation of the manuscript.

==============================
Over the last decades antibiotic resistance has evolved and spread in many different bacterial species. From an evolutionary perspective, drug resistance is an adaptation to a new environment. Population genetic theory predicts that recent adaptations are associated with reduced diversity in the part of the population that carries the adaptive trait, due to a process known as selective sweeps. It is not known if this prediction holds for antibiotic resistance in bacterial species that infect humans. Here, we use a convenience sample of eight datasets covering three species of bacteria (Escherichia coli, Staphylococcus aureus and Enterococcus faecium). The eight datasets include multi locus sequence type information and antibiotic resistance phenotypes for between 53 to 1,094 patient samples and between three and 19 different antibiotics. Consistent with selective sweep theory, we find that, in most cases, sequence-type diversity amongst antibiotic-resistant bacterial strains is lower than amongst antibiotic-susceptible strains. Specifically, in 45 out of 59 comparisons (76%), resistant strains were significantly less diverse than susceptible strains. We also observe that while diversity is lower amongst resistant strains, in all cases there were at least several different sequence types amongst the resistant strains, which suggests that we are dealing here with soft sweeps rather than hard sweeps. Finally, we show that sequence-type diversity of antibiotic-resistant strains is lower if resistance is rare, compared to when resistance is common, which is consistent with rare resistance being due to fewer evolutionary origins. To summarize, our results show that for several different bacterial species, diversity of resistant strains is generally reduced, which is consistent with the evolution of resistance driven by selective sweeps stemming from a limited number of evolutionary origins. In future studies, more detailed analysis of such sweep signatures is warranted.

Introduction

Antimicrobials are drugs that help us combat diseases caused by pathogens that infect some bacteria, viruses, parasites, and fungi. An increase of antimicrobial resistance threatens public health and well-being. According to the World Bank, an additional 7 to 24 million people will be added into the extreme poverty category by 2030 as an impact of antimicrobial resistance (Jonas et al., 2017). In the US alone there are estimated to be 2.8 million antibiotic-resistant infections and 35,000 deaths each year (CDC, 2020). For these reasons, antibiotic-resistant bacterial infections are receiving a lot of attention (Balsalobre, Dropa & Matté, 2014; Barbosa & Levy, 2000; Pendleton, Gorman & Gilmore, 2013).

Portions of this text were previously published as part of a preprint https://www.biorxiv.org/content/10.1101/2022.11.23.517742v1.full.pdf.

While antibiotic resistance is a big problem worldwide, the population genetics of drug resistance evolution outside of the laboratory is not well understood. If we knew more about the origins of antibiotic-resistant bacteria and how they spread, this could help us prevent antibiotic-resistant infections. Several important studies have determined the origins of drug-resistant strains. For example, Enright et al. (2002) showed that Staphylococcus aureus has acquired the SCCmec element at least 11 times. Another S. aureus study focused only on sequence type 5 and showed that the SCCmec element was imported into sequence type 5 S. aureus strains at least 23 times (Nubel et al., 2008). The same study also reported that most of the resulting MRSA strains remain local in one or a few countries (Nubel et al., 2008). Croucher et al. (2014) showed that a soft sweep with multiple origins of drug resistance and vaccine escape happened in Streptococcus pneumoniae. In addition to multiple origins of resistant strains, it has been shown that in most cases, susceptible strains co-exist with resistant strains (Austin, Kristinsson & Anderson, 1999; Blanquart, Lehtinen & Fraser, 2017; Colijn et al., 2009). Co-existence of resistant and susceptible strains may be due to fitness costs of resistance (Andersson & Hughes, 2010; Melnyk, Wong & Kassen, 2015; Pennings, 2023). These results suggest that antibiotic resistance evolution may be best described by local and incomplete soft sweeps from multiple origins.

For many antibiotics, resistance has become more common over the last few decades, in part because of the successful spread of specific resistant strains (Cookson, 2011; Gladstone et al., 2021; Harris et al., 2010). While historically the term “selective sweep” was used for the spread of alleles in recombining eukaryotic species, here we consider the successful spread of a specific resistant strain (i.e., the descendants of a bacterium that acquired resistance) a selective sweep (Croucher et al., 2014; Wilson et al., 2016; Wolff & Garud, 2023). If the rise of resistance in a species or population is due to a single origin of resistance, it would be referred to as a “hard sweep” but if it is due to multiple origins, it would be a “soft sweep” (Hermisson & Pennings, 2017; Messer & Petrov, 2013).

If the number of different origins of antibiotic-resistant strains is fairly small, one may expect that resistant strains are overall less diverse and have fewer sequence types than susceptible strains due to a process known as a selective sweep (Hermisson & Pennings, 2017; Maynard Smith & Haigh, 1974). However, if the number of different origins of antibiotic-resistant strains is large, as suggested by Nubel et al. (2008), diversity of resistant strains may be as high as diversity of susceptible strains. Note that we assume here that sequence types are older than the evolution of resistance, though we realize that may not be true in all cases (Ny et al., 2019). We set out to test for different bacterial species and different antibiotics, whether resistant strains are indeed less diverse than susceptible strains.

In this article we analyze resistance to 38 antibiotics found in six different published studies (eight datasets) focusing on three pathogens: E. coli, S. aureus and E. faecium. For each study and each antibiotic (59 comparisons in total), we determine whether the resistant strains show lower sequence type diversity compared to the susceptible strains as measured by the Gini-Simpson Index, the Inverse Simpson Index, and the Shannon’s Diversity Index (H’) (Jost, 2006; Morris et al., 2014; Simpson, 1949).

With the collected data, we find three main results. First, we find that resistant strains are generally less diverse than susceptible strains. Second, we observe that in all cases there were at least several different sequence types amongst the resistant strains, which suggests that resistance has been acquired multiple times and we are dealing here with soft sweeps rather than hard sweeps. Finally, we find that diversity of resistant strains is affected by how rare the specific resistance is—rarer resistance is associated with lower diversity.

Methods

Data collection

We used data from a convenience sample of published papers that reported antibiotic resistance and multi-locus sequence typing (MLST) for individual samples of bacterial infections. The published articles were found through the NCBI, San Francisco State University (SFSU), and Google Scholar databases over several years of working on this type of data. The main criteria for inclusion of the data were that the article includes both resistant and susceptible strains for each of the antibiotics, mentions all the antibiotics used, and reported MLST information (sequence types) for each patient sample. Articles were excluded if they only reported data for resistant samples or if they only reported summary statistics, but not the information for each sample.

Multi-locus sequence types are commonly used to characterize bacterial strains based on the sequence of a small number of standard housekeeping genes. A combination of each unique allele of the housekeeping genes is assigned a ST number such as ST3, ST15, ST131 and so on (Maiden et al., 1998; Adiri, Gophna & Ron, 2003; Enright et al., 2000; Homan et al., 2002).

Each dataset we collected contains a list of samples (one sample per patient), and for each sample we know whether it is resistant or susceptible to a list of antibiotics. For each antibiotic, we can therefore split a dataset in a population of resistant samples and a population of susceptible samples. Because we also know the multi-locus sequence typing for each sample, we can calculate the sequence type diversity of the population of resistant samples and the sequence type diversity of the population of susceptible samples (using different measures of diversity). The focus of the study is to determine whether the resistant populations are less diverse than the susceptible populations.

In total we collected data from six published articles. Three of these articles reported data on E. coli infections (Yamaji et al., 2018; Adams-Sapper et al., 2013; Kallonen et al., 2017) and all three of these studies were based on samples from surveillance efforts. Two papers are based on the study of S. aureus infections (Wurster et al., 2018; Manara et al., 2018) and one article is based on E. faecium infections (Galloway-Peña et al., 2009). The S. aureus and E. faecium data should be considered convenience samples, as they were not from planned surveillance efforts. However, we still believe that they will give us good estimates of sequence type diversity because none of them were focused on a specific drug resistance outbreak situation. The Yamaji, Adams-Sapper and Wurster articles all treated “intermediate” samples as “resistant”. For the other studies, we excluded samples that were listed as “intermediate.” The Yamaji E. coli article includes two datasets which are collected from the same location, but 17 years apart, therefore we consider it as two different datasets for our analysis (Yamaji et al., 2018). The Kallonen E. coli article includes two datasets, collected from national and local sites, which we consider as two different datasets for our analysis (Kallonen et al., 2017). In total, eight different datasets from the six published papers were collected for the analysis (Table 1).

Table 1 An overview of all the datasets included in the study with information on the antibiotics that are investigated in each dataset.

	Dataset Names	Yamaji 1999	Yamaji 2016	Addams-sapper	Kallonen BSAC	Kallonen CUH	Wurster	Manara	Galloway	
Antibiotic Classification	References	Yamaji et al. (2018)	Yamaji et al. (2018)	Adams-Sapper et al. (2013)	Kallonen et al. (2017)	Kallonen et al. (2017)	Wurster et al. (2018)	Manara et al. (2018)	Galloway-Peña et al. (2009)	
			Pathogen	E. coli	E. coli	E. coli	E. coli	E. coli	S. aureus	S. aureus	E. faecium	
			Number of isolates	225	233	246	1094	415	262	184	53	
			PMID	29436416	29436416	23147723	28720578	28720578	30521630	30424799	19821720	
Beta-Lactams	Penicillin	Amoxicillin				X					
		Ampicillin	X	X			X			X	
		Oxacillin						X	X		
		Penicillin						X	X		
Cephalosporins	1st	Cefalotin					X				
Cephalosporins-BS			X						
2nd	Cefoxitin					X				
Cefuroxime				X	X				
Cefuroxime axetil					X				
3rd	Cefotaxime				X	X				
Ceftazidime				X	X				
4th	Cefepime			X		X				
Monobactams		Aztreonam			X		X				
Carbapenems		Carbapenem			X						
		Ertapenem					X				
		Imipenem				X					
		Meropenem					X				
DNA Topoisomerases	Quinolones	Ciprofloxacin	X	X		X	X				
		Fluoroquinolones			X						
		Levofloxacin						X	X		
Protein synthesis	Aminoglycosides	Amikacin					X				
		Gentamicin				X	X	X	X	X	
		Streptomycin								X	
		Tobramycin					X				
Oxazolidinones		Linezolid						X	X		
Lincosamides		Clindamycin						X	X		
Macrolides		Erythromycin						X			
Tetracyclines	1st	Tetracycline						X	X		
3rd	Tigecycline				X	X		X		
Combination	Penicillin Combinations	Amoxicillin-Clavulanic acid				X	X				
		Piperacillin-Tazobactam				X	X				
Sulfonamides combinations	Trimethoprim-Sulfamethoxazole	X	X	X		X	X	X		
Others	Lipopeptides	Daptomycin							X		
Fusidane	Fusidic acid							X		
Rifamycins	Rifampicin							X		
Diaminopyrimidines	Trimethoprim					X				
Glycopeptide	Teicoplanin							X		
			Vancomycin						X	X	X	

Escherichia coli datasets

For E. coli, there are five datasets from three different published articles:

1. Yamaji_1999 consists of 225 isolates from patients with urinary tract infections (UTIs) collected between 1999 and 2000 at the University of California, Berkeley, California, USA. Resistance phenotypes are available for three different antibiotics (ampicillin, ciprofloxacin and trimethoprim-sulfamethoxazole).

2. Yamaji_2016 consists of 233 isolates from patients with urinary tract infections (UTIs) collected between 2016 and 2017 at the University of California, Berkeley, California, USA. Resistance phenotypes are available for three different antibiotics (ampicillin, ciprofloxacin and trimethoprim-sulfamethoxazole).

3. Addams-Sapper consists of 246 isolates from patients with bloodstream infections collected between 2007 and 2010 at San Francisco General Hospital (SFGH), California, USA. Resistance phenotypes are available for six different antibiotics (cefepime, aztreonam, cephalosporins-BS, fluoroquinolones, and trimethoprim-sulfamethoxazole).

4. Kallonen_BSAC consists of 1,094 isolates from patients with bacteremia collected between the years of 2001 and 2011 by the British Society for Antimicrobial Chemotherapy (BSAC) from 11 hospitals across England. Resistance phenotypes are available for 10 different antibiotics (amoxicillin, amoxicillin-clavulanic acid, cefotaxime, ceftazidime, cefuroxime, ciprofloxacin, gentamicin, imipenem, piperacillin-tazobactam, and tigecycline).

5. Kallonen_CUH consists of 415 isolates from patients with bacteremia collected between the years of 2006 and 2012 from the local diagnostic laboratory at the Cambridge University Hospitals in England. Resistance phenotypes are available for 19 antibiotics (amikacin, amoxicillin-clavulanic acid, ampicillin, aztreonam, cefalotin, cefepime, cefotaxime, cefoxitin, ceftazidime, cefuroxime, cefuroxime axetil, ciprofloxacin, ertapenem, gentamicin, meropenem, piperacillin-tazobactam, tigecycline, tobramycin, and trimethoprim)

Staphylococcus aureus datasets

There were two different datasets for S. aureus from two different published articles:

5. Wurster consists of 262 clinical isolates from ocular and otolaryngology infections collected from January to December 2014 from Massachusetts Eye and Ear, a Harvard teaching hospital in Boston, Massachusetts, USA. Resistance phenotypes are available for 11 antibiotics (clindamycin, erythromycin, gentamicin, levofloxacin, linezolid, oxacillin, penicillin, tetracycline, trimethoprim-sulfamethoxazole, and vancomycin).

6. Manara consists of 184 clinical isolates from respiratory tract infections (RTIs), soft tissue and skin lesions of patients collected from 2013 to 2015 at the Anne Meyer’s Children’s University Hospital, Florence, Italy. Resistance phenotypes are available for 14 antibiotics (clindamycin, daptomycin, fusidic acid, gentamicin, levofloxacin, linezolid, oxacillin, penicillin, rifampicin, teicoplanin, tetracycline, tigecycline, trimethoprim-sulfamethoxazole, and vancomycin).

Enterococcus faecium dataset

There is one dataset with Enterococcus faecium samples:

7. Galloway consists of 53 clinical isolates from nosocomial patients collected from 1971 to 1994 at diverse geographic locations in the United States. Resistance phenotypes are available for 4 antibiotics (ampicillin, gentamicin, streptomycin, and vancomycin).

Data preparation

Custom R scripts were used to prepare the data from the different sources for analysis (R Core Team, 2021). Specifically, for each dataset, we created a .csv file which contains information of each sequence type, each drug, the number of resistant samples and the number or susceptible samples. Before analysis of the data, we removed any sequence types that were marked as “minor”, “ND” or “-” because these categories would likely include multiple different sequence types. Table 2 shows an example dataset (the smallest of the datasets) for illustration.

Data accessibility

All the data and R scripts are available on GitHub (Antibiotic_Resistance_Data_Analysis.git).

Data analysis

This study is based on eight different datasets from six different published articles. Because antibiotic resistance is likely a newly evolved trait for the bacterial species, the expectation is that sequence type diversity is lower among resistant populations when they are compared to the susceptible ones.

Several approaches for the analysis of the datasets to study the antibiotic diversity among resistant and susceptible populations were taken. For each dataset and each antibiotic, the samples were split into resistant and susceptible samples. Next, we calculated diversity (using three different indices) for resistant and susceptible populations for each antibiotic in each dataset. If the diversity of resistant populations was lower than for the susceptible population, we used a bootstrapping approach to determine if this difference was significant. Finally, we analyze observed levels of diversity using a linear model to determine whether the proportion of resistant strains can explain the level of diversity amongst resistant strains.

Table 2 Example dataset.

We show here Yamaji_1999, which is the smallest of the datasets.

Sequence type	Antibiotic	Number of resistant samples	Number of susceptible samples	
ST 95	Ampicillin	1	33	
ST 127	Ampicillin	3	21	
ST 73	Ampicillin	6	17	
ST 69	Ampicillin	19	7	
ST 131	Ampicillin	2	5	
ST 10	Ampicillin	2	9	
ST 95	Trimethoprim-Sulfamethoxazole	1	33	
ST 127	Trimethoprim-Sulfamethoxazole	1	23	
ST 73	Trimethoprim-Sulfamethoxazole	1	22	
ST 69	Trimethoprim-Sulfamethoxazole	17	9	
ST 131	Trimethoprim-Sulfamethoxazole	1	6	
ST 10	Trimethoprim-Sulfamethoxazole	2	9	

Measures of diversity

We used three common measures of diversity: Gini-Simpson Index, Inverse Simpson Index, and Shannon’s Diversity Index (H’).

The Gini-Simpson Index and the Inverse Simpson Index are popular indices to measure diversity, and they are often used to quantify biodiversity. The Gini-Simpson Index is calculated as 1- ∑(pi)2 where pi is the proportional abundance of the ith sequence type. The Gini-Simpson Index ranges from zero to one, where 0 represents no diversity and 1 represents the highest diversity. The Inverse Simpson Index is calculated by 1 / ∑(pi)2 and the index values can be higher than one. Another popular diversity index is the Shannon’s Diversity Index (H’), which is calculated as—∑pi ln(pi). Shannon’s Diversity Index measures the uncertainty in predicting the sequence type of the identity of sequence type samples that are randomly taken from the datasets. Shannon’s Diversity Index values can be higher than one. For the three measures of diversity indices, the higher the indices values, the higher the diversity for the resistant or susceptible population (Jost, 2006; Morris et al., 2014; Simpson, 1949).

Analysis of significance for resistant and susceptible antibiotic populations

After calculating the diversity of the resistant populations and the susceptible populations for each drug in each dataset, a bootstrapping approach was used to test our hypothesis (Kulesa et al., 2015). For each antibiotic in each dataset, a randomized set of resistant and susceptible populations were simulated 1,000 times by resampling from the entire population with replacement. The sample sizes for these simulated populations were the same as the observed resistant and susceptible populations. For example, there were 33 Ampicillin-resistant samples in the Yamaji_1999 dataset and 92 Ampicillin-susceptible samples; the bootstrapping procedure created populations of these sizes by sampling with replacement from the entire dataset consisting of 92 + 33 = 125 strains. Diversity of simulated resistant and susceptible populations were determined by calculating diversity values for these randomized groups. We then determined how often the difference in diversity between the simulated resistant and susceptible populations was equal to or larger than the observed difference in diversity between the resistant and susceptible populations. If this was the case in less than 5% of the cases, we considered the difference significant. Reported p-values are not Bonferroni or otherwise corrected.

Regression analysis: what determines diversity

We were interested to better understand which factors determine diversity of resistant populations. While our dataset is not extensive enough to test for many different factors, a linear model was used to determine whether the proportion of resistant strains in populations (i.e., what fraction of patients in a dataset carried resistance to a specific antibiotic) has an effect on observed diversity. However, to do this, we have to use a normalized version of the Gini-Simpson Index (GSI): GSI_nor = GSI / GSI_max.

The reason why we need to use a normalized version is that when there are only a few patients with resistant strains, the diversity cannot be as high as when there are many patients with resistant strains. For example, if there are just two patients with resistant strains, we can have at most two sequence types, with each a frequency of 50%. In that case the maximum possible value of the Gini-Simpson Index is 1-(0.52 +0.52) = 0.5. Therefore, even if resistant and susceptible labels were randomly assigned to the patients, the findings of the fraction of patients with resistant strains would still show an effect on diversity. To deal with this issue, we decided to use a normalized version of Gini-Simpson Index by dividing the observed Gini-Simpson Index value by the maximum possible value of the Gini-Simpson Index given the sample size, similar to an approach taken by Garud & Rosenberg (2015). We find that, as we hoped, when we randomly assign labels (resistant and susceptible), the normalized Gini-Simpson Index values do not go up with the fraction of resistant strains (code and results in GitHub repository).

To determine if the fraction of patients with resistant strains has an effect on the observed normalized Gini-Simpson Index (GSI) values, we used a linear model in R using the following formula: lm(formula = GSI_nor ∼FracRes + Dataset, data = Data). Here we include “Dataset” as a fixed effect because diversity values may vary widely between the different datasets as they are from different species, locations and years.

Results

Examples of sequence type diversity in susceptible and resistant populations

For each of the antibiotics in each of the datasets we calculated the diversity for resistant and susceptible populations and used bootstrapping to determine whether resistant samples were significantly less diverse than susceptible samples. In Fig. 1, we visualize the results for the Wurster dataset (resistance to penicillin and oxacillin) and the Kallonen_CUH dataset (tobramycin resistance). In the Wurster dataset, the Gini-Simpson Index is 0.9 for penicillin-susceptible strains and 0.91 for penicillin-resistant strains (Fig. 1A), this difference is not significant. The pie charts (Figs. 1B and 1C) also illustrate similar and high sequence type diversity among penicillin-resistant and penicillin-susceptible strains. Also, in the Wurster dataset, the Gini-Simpson Index is 0.93 for oxacillin-susceptible strains and 0.73 for oxacillin-resistant strains (Fig. 1D), which is a significant difference (p < 0.001). This difference is also visible in the pie charts (Figs. 1E and 1F). In the Kallonen_CUH dataset, the Gini-Simpson Index is 0.93 for tobramycin-susceptible strains and 0.47 for tobramycin-resistant strains (Fig. 1G), which is a significant difference (p < 0.001). This difference is also visible in the pie charts (Figs. 1H and 1I).

Figure 1 Diversity among susceptible and resistant populations.

Diversity among susceptible and resistant populations. (A) Gini-Simpson Index of diversity among penicillin-susceptible samples (teal) and penicillin-resistant samples (red) for the Wurster dataset. (B) Sequence type abundance among penicillin-susceptible samples in the Wurster dataset. (C) Sequence type abundance among penicillin-resistant samples in the Wurster dataset. (D, E, F) like A, B, C but for oxacillin-susceptible and oxacillin-resistant samples in the Wurster dataset. (G, H, I) like A, B, C but for tobramycin-susceptible and tobramycin-resistant samples in the Kallonen_CUH dataset.

Plots for other antibiotics and all datasets are available in the GitHub repository.

Sequence type diversity for resistant populations is usually lower than for susceptible populations

We calculated three diversity indices for all datasets but will focus here on the Gini-Simpson Index for diversity. We plotted the Gini-Simpson Index for resistant and susceptible strains for each drug for each of the datasets (Fig. 2A) and the fraction of samples that was found to be resistant (Fig. 2B). As can be seen in the figure, in most of the cases, resistant strains were significantly less diverse than susceptible strains.

Figure 2 Diversity values of resistant samples.

(A) Comparison of the Gini-Simpson Index for resistant and susceptible strains in all datasets. Trim-Sulf (trimethoprim-sulfamethoxazole), Pip-Taz (piperacillin-tazobactam), Amoxi-Clav (amoxicillin-clavulanic acid). (B) Resistant fraction for each antibiotic in each dataset.

E. coli datasets. In the Yamaji dataset from 1999–2000, the samples were tested against three antibiotics, but no samples were resistant against ciprofloxacin, so it was not possible to calculate diversity. For ampicillin and trimethoprim-sulfamethoxazole (Trim-Sulf), the resistant populations were significantly less diverse than the susceptible populations, although the difference was bigger and more significant for trimethoprim-sulfamethoxazole. In the later data (2016–2017) from the same study, we see that resistance has gone up for all three drugs (Fig. 2B). In terms of diversity, we see the lowest diversity for the (still rare) ciprofloxacin resistant strains, higher diversity for the more common trimethoprim-sulfamethoxazole resistant strains and even higher diversity for the very common ampicillin-resistant strains. In the latter case, the resistant strains even display more diversity than the susceptible strains.

In the Addams-Sapper dataset resistance was determined for five drugs (cefepime, aztreonam, cephalosporins-BS, fluoroquinolones and trimethoprim-sulfamethoxazole). In all cases except for trimethoprim-sulfamethoxazole, the resistant strains were significantly less diverse than the susceptible strains (Fig. 2A). Trimethoprim-sulfamethoxazole was the most common resistance in this dataset (Fig. 2B).

In the Kallonen-BSAC dataset, resistance was determined for seven drugs (ceftazidime, gentamicin, cefotaxime, cefuroxime, amoxicillin-clavulanic acid and amoxicillin). In all cases except for amoxicillin, the resistant strains were significantly less diverse than the susceptible strains (Fig. 2A) and amoxicillin was the most common resistance in this dataset.

In the second Kallonen dataset (Kallonen-CUH), resistance was determined for 15 drugs (amikacin, cefoxitin, cefepime, ceftazidime, aztreonam, gentamicin, piperacillin-tazobactam, cefotaxime, tobramycin, cefuroxime, cefuroxime axetil, ciprofloxacin, cefalotin, amoxicillin-clavulanic acid and ampicillin). In all cases except amikacin and cefoxitin resistant strains were significantly less diverse than the susceptible strains (Fig. 2A).

S. aureus datasets. In the Wurster dataset, resistance was determined for 8 drugs (trimethoprim-sulfamethoxazole, gentamicin, tetracycline, levofloxacin, oxacillin, clindamycin, erythromycin and penicillin). In four cases resistant strains were significantly less diverse than the susceptible strains (Fig. 2A). In the Manara dataset, resistance was determined for 12 drugs (teicoplanin, daptomycin, linezolid, trimethoprim-sulfamethoxazole, fusidic acid, rifampicin, tetracycline, gentamicin, levofloxacin, clindamycin, oxacillin, penicillin). In nine cases resistant strains were significantly less diverse than the susceptible strains (Fig. 2). For both Manara and Wurster, the difference is not significant for the drugs with lowest levels of resistance and those with the highest level of resistance.

E. faecium dataset. In the Galloway-Peña dataset, resistance was determined for four drugs (vancomycin, gentamicin, streptomycin and ampicillin). In two cases (ampicillin and gentamicin) resistant strains were significantly less diverse than the susceptible strains (Fig. 2A). In the other two cases (streptomycin and vancomycin), there was no significant difference in diversity between resistant and susceptible strains (Figs. 2A and 2B).

In total, out of 59 comparisons of the Gini-Simpson Index between resistant and susceptible populations, in 45 cases (76%) we found that resistant populations were significantly less diverse. In eight cases where there was no significant difference, the level of resistance was very low, leading possibly to low power. In five cases resistant populations had higher diversity than susceptible populations: these were ampicillin resistance in E. coli in the Yamaji 2016 dataset, trimethoprim-sulfamethoxazole resistance in E. coli in the Addams-Sapper dataset, amoxicillin resistance in E. coli in the Kallonen BSAC dataset, penicillin resistance in S. aureus in the Wurster dataset and penicillin resistance in S. aureus in the Manara dataset.

Regression analysis for the entire dataset

In the previous sections, we showed that diversity is typically lower for resistant populations than it is for susceptible populations. We noticed that while most comparisons were significantly different (resistant populations less diverse), the cases where the diversity was very high in the resistant population were the cases where resistance was very common. Here we will use a linear model to determine whether there is a significant effect of how common resistance is on the diversity of resistant samples. For this analysis, we worked with the normalized Gini-Simpson Index values (see Methods) to make sure that small numbers of resistant strains don’t bias the results.

We fitted a linear model to the normalized Gini-Simpson Index values, with the fraction of samples that are resistant (FracRes) and the dataset as explanatory variables and found that indeed, diversity goes up with increasing fraction of resistant samples (FracRes), see Fig. 3.

Figure 3 The fraction of samples that is resistant vs the normalized Gini-Simpson Index, with fitted linear model for each dataset.

Discussion

In this study, we used existing clinical data on three different bacterial pathogens (E. coli, S. aureus and E. faecium) to determine whether drug resistant populations are generally less diverse than susceptible populations. Diversity here is considered at the between-host level and assessed by considering multi-locus sequence types. High diversity of resistant populations therefore means that different patients are infected with resistant strains of different multi-locus sequence types, whereas low diversity means that patients are infected with only a limited number of multi-locus sequence types represented among resistant strains.

We found that indeed, in all eight datasets, in most cases, resistant populations are less diverse than susceptible populations. All in all, in 45 of 59 comparisons (76%), the resistant populations were significantly less diverse than susceptible strains (Fig. 2). While this shows that resistant samples consist of fewer sequence types, the number of sequence types that carry resistance is larger than 1 in all cases. This shows that resistance has most likely evolved or been acquired (in the case of HGT) multiple times in every dataset and for every specific antibiotic we consider. Therefore, we can conclude that at the host population level, the evolution of antibiotic resistance typically occurs by multi-origin soft selective sweeps (Hermisson & Pennings, 2017; Messer & Petrov, 2013). This had previously been shown in a few other cases (Croucher et al., 2014; Nubel et al., 2008; Wilson et al., 2016), but we are not aware of another study that looked at multiple pathogens and multiple datasets.

Consistent with the theory of soft and hard selective sweeps (Hermisson & Pennings, 2017; Messer & Petrov, 2013), if resistance evolution or acquisition happens only rarely, we expect only few different sequence types to harbor drug resistance, which leads to low diversity of resistant strains. If resistance evolution or acquisition is common, for example, when horizontal gene transfer (HGT) happens often, we expect many different sequence types to harbor drug resistance, leading to high diversity. This could lead to a situation where high mutation or HGT rates would lead to resistance being common and resistance populations being diverse. Alternatively, in cases where HGT plays a role, we could expect that when resistance becomes common over time, it could be transferred to other sequence types within the same species, which will increase the sequence type diversity of resistant strains over time. Thus, there are at least two different models that could lead to common resistance being associated with high diversity. We found that indeed, there was a significant effect of the fraction of samples that was resistant in a dataset for a given antibiotic on the diversity of these resistant strains (Fig. 3).

The Yamaji datasets came from the same location (Northern California) but 17 years apart. In the first dataset (Yamaji_1999), trimethoprim-sulfamethoxazole resistant and ampicillin resistant strains are significantly less diverse than their susceptible counterparts. 17 years later (Yamaji_2016), diversity of the resistant strains has gone up in both cases. In addition, for ampicillin, the frequency of resistance has gone up (Fig. 2A) and there is no longer a difference in diversity between resistant and susceptible strains. While this is a small dataset and the only case where we have longitudinal data, it suggests that over time, drug resistant strains can become both more common and more diverse.

A limitation of this study is that we had only eight datasets to work with. In addition, we had only multi-locus sequence type data which could be used to quantify diversity. We did not have genome sequences which would have allowed more precise study of how often and when resistance evolved in these species. Here we worked with the assumption that sequence types are old enough to be considered immutable, whereas resistance evolution is more recent and happens on the background of the immutable sequence types. A study using phylogenetic or clustering approaches based on whole genomes would allow for more precise analysis.

In some cases, the same genetic mechanism leads to resistance to multiple drugs (e.g., a plasmid with several resistance genes), but we did not take that into account in our analysis. This means that some of the 59 comparisons we tested should not be considered independent comparisons. However, because our results are quite similar between the different datasets and even different species of bacteria, we believe that our findings could indeed be quite general. Another data availability issue we ran into was the fact that some studies listed a small number of sequence types as “other” or “minor”. We removed these sequence types from our analysis. For most of the datasets, this is not a big issue (e.g., 1 sample out of 53 for the Galloway dataset, 11 samples out of 184 for the Manara dataset), but for the Yamaji datasets, it affects a large number of samples (100 out of 224 (44%) in Yamaji 1999 and 85 out of 233 (36%) for Yamaji 2016). It is unfortunate that data is not always completely reported. In addition, many other datasets we considered did not include patient-level sequence type data at all. We are hopeful that the recent push for data sharing will make this type of analysis more feasible in the future.

Despite using old-fashioned data (multi-locus sequence types and not whole-genome sequences), we believe that this study will be useful for the field and could inspire follow-up studies. It would be interesting to test whether these patterns are also seen when whole-genome sequencing data is used. We also could test if these patterns are seen in global samples as opposed to local samples (sweeps may be so soft at the global level that we can’t see an effect on diversity anymore) (Ralph & Coop, 2010). We could determine how these patterns change over time (more recent adaptation is associated with softer or harder sweeps?) and whether the mode of adaptation (horizontal gene transfer vs chromosomal mutations or both) influences the results. Finally, an important question pertains to how selective sweeps interact with apparent stabilizing selection (resulting in coexistence of resistance and susceptibility) (Austin, Kristinsson & Anderson, 1999; Blanquart, Lehtinen & Fraser, 2017; Colijn et al., 2009; Krieger et al., 2020; Pennings, 2023).

With this study, we aim to bridge the world of population genetics, evolution and selective sweeps with the study of clinical bacterial samples and antibiotic resistance. A lot of effort goes into fighting antibiotic-resistant infections. We believe that a better understanding of the evolution and spread of antibiotic-resistant strains will ultimately contribute to finding better ways to reduce the number of antibiotic-resistant infections.

We would like to thank the SF BUILD writing retreat (August 15-18, 2022, Westerbeke Ranch Conference) for giving us time and space to write this article. We thank Dr Alison Feder and Dr Nandita Garud for discussions.

Additional Information and Declarations

Competing Interests

Author Contributions

Data Availability

The authors declare there are no competing interests.

Anjani Pradhananga analyzed the data, prepared figures and/or tables, authored or reviewed drafts of the article, and approved the final draft.

Lorena Benitez-Rivera analyzed the data, prepared figures and/or tables, authored or reviewed drafts of the article, and approved the final draft.

Candace Clark analyzed the data, prepared figures and/or tables, and approved the final draft.

Kaho H. Tisthammer analyzed the data, prepared figures and/or tables, and approved the final draft.

Pleuni S. Pennings conceived and designed the experiments, analyzed the data, prepared figures and/or tables, authored or reviewed drafts of the article, and approved the final draft.

The following information was supplied regarding data availability:

The data is available at GitHub and Zenodo:

-https://github.com/AntibioticResistance/Antibiotic_Resistance_Data_Analysis/releases/tag/1.0.

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
