# Peer review of "Evidence for sweep signatures in antibiotic-resistant strains in three species of bacteria"

_PeerJ, doi:10.7717/peerj.18178_

## Round 0.1 · original submission · Major Revisions

While all of the reviewers agreed that the manuscript was well written, there was some disagreement amongst the reviewers as to the validity of the findings. You should address the comments of all reviewers.

·

Basic reporting

In this brief article, the authors use published data to investigate whether antibiotic resistance is associated with less genetic diversity among bacteria, as predicted by population genetic theory on selective sweeps. The manuscript is clearly written, well-situated in the literature, and conforms to PeerJ and disciplinary norms. The figures are generally clear (but see detailed comments below). The raw data and code are available in GitHub. I have a few suggestions to improve the clarity of the paper:

1. The second Introduction paragraph mentions how ‘susceptible strains co-exist with resistant strains.’ This could be due to resistant strains suffering fitness costs. Such costs of resistance could be mentioned.

2. Also in the introduction, line 72, you could specify that sweeps still occur in the scenario of highly diverse resistant strains, just that this is a soft sweep of resistance arising on multiple genetic backgrounds. This would help clarify the concept of soft sweeps.

3. Introduction line 79: “lower sequence type diversity.” Can you give some information on the time scales of divergence among STs, and whether the time scale is expected to be relevant to the evolution of antibiotic resistance? Even a rough estimate would be useful. There is an implicit assumption throughout the manuscript that resistance generally evolves recently (subsequent to the divergence of STs) but in some cases resistance could be an ancient trait that is ancestral to the entire species, therefore common to all STs (except for occasional loss of resistance). For example, on line 202, could you support the idea that ‘resistance is likely a newly evolved trait’? There are many examples of ancient resistance (e.g. https://doi.org/10.1371/journal.pone.0069533)

4. Figure 2A needs a colour legend. Also the asterisks and ‘n.s.’ text for) is too small defined, and the meaning of these symbols (presumably statistical significance) should be defined in the legend (same for Figure 1).

Experimental design

The article describes original research with a well-defined and relevant question. It is clearly stated how this work fills a knowledge gap in our understanding of the evolution of antibiotic resistant bacteria.

The methods are generally well defined, but I have two specific suggestions to improve clarity and reproducibility:

1. Were specific search terms used for the literature review? This could be mentioned in the first Methods paragraph.

2. In the linear model (lines 266-269), could you specify whether ‘dataset’ is included as a fixed or random effect? I think it is fixed, but please clarify.

Validity of the findings

In general, the conclusions are well-supported by the data and analyses. The consistent results across studies provides good support for the generality of the association between resistance and lower ST diversity, and certain caveats are appropriately mentioned in the discussion. I would like to raise a few additional caveats that could be addressed or mentioned to further support the conclusions.

1. One important caveat to consider is that not all STs are equally related: some could represent ancient evolutionary divergences and others more recent ones. In some cases, it should be possible to cluster STs (e.g. using the BURST algorithm or other tools described on pubmlst.org) which could provide additional resolution to the study – e.g. does one ST likely descend from another, such that resistance in both STs constitutes one, not two, independent acquisitions of resistance? Even if it is not feasible to conduct such analyses, it would be useful to comment on these considerations in the discussion.

2. A related point is that there are likely biases in how STs and sensitive/resistance isolates are sampled. For example, sampling might be focused on a hospital outbreak of a resistance clone, which would bias the dataset toward a large number of isolates from the same resistant ST. I appreciate the permutation tests used to control for different sample sizes of resistant vs. sensitive isolates, but I believe the permutations still do not account for this particular type of sampling bias. I am not sure if there is an easy way to control for this – although see my point directly above: could the relatedness among STs be incorporated into the permutation test? If not, or if this proves too difficult, it could still be mentioned in the discussion if and how such sampling bias might impact the results. Given the relatively small number of studies included in the analysis, the individual publications might give clues about whether or not a hospital outbreak (for example) or other likely clonal expansion of a resistant clone was targeted for sampling.

3. It is interesting that ‘the cases where the diversity was very high in the resistant population were the cases where resistance was very common’ (lines 368-370). To further support the idea that resistance increases over time (in both the total proportion of resistance bacteria and the diversity of resistant STs), would it be possible to use information about the dates when each antibiotic came into clinical use? One would expect the first antibiotics to enter clinical use (e.g .1940s and 1950s) would select for more resistant isolates on more STs compared to more recent antibiotics (1960s…). Just an idea if this information is available.

4. In the Discussion, it is rather a big jump on line 398 that “we can conclude that [...] resistance typically occurs by multi-origin soft selective sweeps.” In cases where resistance is common, is it possible that resistance is an ancestral trait of the species (see my comment above)? Sweeps imply that there is an increase in resistance-conferring alleles over time, driven by positive selection. You actually have some interesting evidence for this, which is mentioned later in the Discussion (lines 420-428) but perhaps would be more useful to bring in earlier to support the evidence for sweeps.

·

Basic reporting

This paper is very well written, and care is taken to explain the context and methods in detail. Figures are very clear and well presented. Data and scripts are made available via GitHub.

In lines 132-161 the authors summarise the input studies. I would like to request that the authors include in this (or add after) a statement or statements clarifying the nature of sampling. I understand the input studies to be census-style samples over a period of time at a particular (usually hospital) location. This is important to reassure the reader that there is not a filter that is being applied that might be associated with resistance/susceptibility status. (Elsewhere the authors are clear on discarding ambiguous sequence types, although it might also be helpful to review input studies for handling of intermediate susceptibility).

It is possible that ascertainment biases (affecting proportions of resistant versus susceptible strains) may operate if clinical entailments of resistance influence the likelihood of hospital follow-up from primary care. This reviewer cannot think of a negative impact of this on the study conclusions, however.

I have reviewed the code – in particular Diversity_Indices_WithFunctions.R. Please note that the drugRfiles object is not defined but appears to be the same as ResDocs object – please could authors amend this in the code.

Referencing is thorough, but I have two minor points related to Jonas et al. (2017). First would the authors mind updating the World Bank URL for stability? Second, I note that the 24 million estimate of people added to the extreme poverty category is an outcome of the report’s “high-AMR impact” category. Since they also offer a “low-AMR impact” simulation scenario/simulation it would be good to add this rider to reflect uncertainty in findings.

In line 289 – may I ask whether additional plots should be placed in a supplementary section in addition to the GitHub repository? Based on my review of GitHub content, the plots that were chosen for the principal manuscript are representative.

Some typographical corrections follow:
Full stop missing in line 46.
Sentence beginning line 60 is incomplete – suggest “that a soft sweep [occurred] with multiple…”
Revise bracket use in line 71
Line 95 – “the paper include” > “a paper includes” – and subsequent verbs
Line 243 – stray hyphen to delete
Line 262 – bracket formatting
Line 294 – reference made to labelling in figure 1 – does this mean shaded? Individual STs do not appear to be labelled in this plot.
Line 304 – typo: “resistant strains were significantly less diverse than [susceptible] strains”

Very minor: it may be helpful to clarify at the outset that the article is referring to incomplete selective sweeps (although this is mentioned in the body of the manuscript). It is possible that some readers may expect this to refer to variation after fixation.

Experimental design

A key estimand in this study is diversity (contrasting antibiotic resistant and susceptible strains). First it is proposed that events in the world may lead to reduced real diversity among resistant strains (= the estimand). These events are the emergence or acquisition (by HGT) of resistance in a subset of the population (in a time-dependent manner). Because this occurs in a subset of the population it is essentially a sampling process. Second the authors also note that the smaller proportion of resistant strains in a sample/dataset can result in reduced measured diversity (= the estimate). This is also a sampling process, but it is artefactual.

The manuscript shows the differences in measured/estimated diversity from datasets illustrating (mostly) lower diversity among resistant strains (figure 2) and then examine the relationship between the proportion of samples that are resistant and diversity normalized by the maximum possible measured diversity (figure 3). This normalization process is an effort to correct for the second process and arrive at a reliable estimate of diversity.

Since, for most antibiotics, the resistant proportion is < 0.5 (figure 2B), one might suppose that the second (artefactual) sampling process would affect interpretation of figure 2A. One might consider an additional panel with normalized GSI values, but I am satisfied with the stated boostrapping approach (described in lines 231-243 and instantiated in Diversity_Indices_WithFunctions.R). I note that the simulations 1. pick with replacement and 2. pick resistant and susceptible data without dependency. In other words, they are essentially identically conditioned multinomial picks varied only by size of sample. The authors may wish to comment briefly on this (low priority).

I do have a request regarding the relational analysis with normalized data (figure 3). In lines 262-264 simulated data is described for testing the linear model. Please would the authors share this simulated data (or code and summaries thereof)? May I ask for the text here to be expanded to report more detail: can you describe a distribution of p values as well as a distribution of effects (which may include negatives)?

Let me justify this starting with the intuition. Taking the example described in lines 254-256 of the submission the authors report that the maximum possible value of the Gini-Simpson Index (GSI) is reduced in a sample of n=2 to 0.5 (and this is then used for normalization). However, what is also reduced is the likelihood of observing 2 different types if the population from which the sample is taken contains some fixed number of types. For example, if a population contained 50% type A and 50% type B, a sample of size 2 would have a 50% chance of estimating a GSI of 0, while successively larger samples would be more likely to provide the correct estimate of 0.5 or one that is close to it. This is relevant to the testing of significance as well as to the “low power” comment on line 359.

To follow up on this intuition I have simulated the expected effect of sampling from a population with a fixed composition (and therefore a “real” diversity; see attachments). This supports my intuition that, while normalization corrects the expected value of the diversity measure (and brings it in line with the real value plotted as a red dashed line), it also increases the spread of estimates in smaller samples. I would like the authors to comment on this in relation to the observations in figure 3.

In the form of an argument: a concern could be that a left tail + higher variability could lead to some deflation in GSI estimates from resistant, smaller samples, especially if these estimates are not robust to extreme values (as is the median in the box plots). Seeing figure 3 from his perspective may lead to some concern that the regression analysis may be negatively influenced by this. The counter is to provide more detail on the findings from simulated data (as requested above and re lines 262-264) and to describe why the estimated diversity across the datasets is robust.

Validity of the findings

Despite my responses above I do believe that the main findings are robust and that the evidence most likely supports a reduced real diversity among resistant strains. This is because the detailed procedures related in the first analysis (including the empirical p value-based assessment for figure 2B) support this contention. I would be happy to see a little more detail on the simulation supporting the analysis in figure 3 (as outlined in the experimental design section) to strengthen this conclusion.

Additional comments

I commend the authors for their very clear exposition throughout the manuscript (also extending to the code base which is highly readable). I have enjoyed reviewing this manuscript and consider it to be a valuable contribution.

Reviewer 3 ·

Basic reporting

The article is well written and easy understand. The data are publicly available and script are shared. However I see major problems in the logic and the analyses. The authors hypothesized that bacterial strains that present resistance to antibiotics are less genetically diverse than the one susceptible. They assume that this is due to select sweep. First, the literature lacks important studies that looked at how adaptation works in bacteria (see Richard Lenski's lab experiment for instance). Second, author are referring to selective sweep all along the manuscript but they actually should refer to clonal interference. Selective sweep is positive selection acting on a particular allele of a gene that will sweep neutral variation associated with it to get to fixation. However, the in the case of this study, the authors just look at MLST genes which are not the genes associated with antibiotic resistance. So lack of diversity could be due to the bottleneck endured after applying antibiotics and not related to the spread of the resistance. So the question that this study tried to answer is more : is the diversity of strains harboring resistance lower than the one of susceptible strains.

Experimental design

In my opinion, there are problems with the methods and how they answered the question. First, they do not show a measure of genetic diversity for within species comparison but a measure of diversity used normally for between species comparison. A classical statistics to measure within species genetic diversity is pi or theta. To do so it is necessary to recover the sequences, align them and estimate those quantity. This shall be done as you might expect differences in result because some alleles are shared between types. Second, it is essential to know if the resistance is encoded by a plasmid or by the chromosome. You expect radical differences in how the resistance will spread and how it will affect nuclear genes diversity. Then separate test should be performed accordingly. Third, the statistics should be redone and more elaborate. You can fit a linear mixed model that will have diversity as a dependent variable, resistance status as independent variable and Dataset as random variable. I think bootstrap approach is unnecessary in your case and that your glm (it is actually a linear model you do not generalized ie your dependent variable is supposed to be normal) is incorrect as your residuals are not randomly distributed (figure 4). Linear model are anyway incorrect on proportion data.

Validity of the findings

I do not think the results support the findings. The authors should redo the analysis and clearly assess what they want to show. If they want to look at selective sweep, they should analyse sequence data and not just strain. They should also be careful that plasmid encoded resistance will not lead to selective sweep but just the spread of the plasmid within population without drastic reduction of diversity.
Finally, there is also a cost to resistance. It has been shown that resistance are lost fast as soon as no more selective pressure is applied through the exposition to antibiotics. So basically your susceptible strains can be one that have lost resistance and spreading in the population. So you mix ancestral populations, selected populations for antibiotics and populations under relaxed selection because some antibiotics are no more in use. That is problematic in the way you interpret your data. In conclusion, the authors have to reassess the way they analyzed the data, the objective of their study and what they conclude from it.

---

## Round 0.2 · Minor Revisions

Both reviewers were pleased that their previous concerns had been addressed, and it is evident that you have tried to improve the manuscript.

Reviewer 2 had some minor comments to still be addressed.

In addition, you reference Prof. Enright's work in the manuscript, which led me to recall that his first PhD student (Alasdair Monk) had a similar (but less involved) analysis of Simpson's Index of Diversity values for STs between MRSA and MSSA strains in his thesis (but I don't think it was ever published). Might be worth looking at if you can find a copy online (from University of Bath). Don't worry if you can't though - it's not essential. Just thought it might be interesting.

·

Basic reporting

No comment. All my previous comments have been addressed.

Experimental design

No comment. All my previous comments have been addressed.

Validity of the findings

No comment. All my previous comments have been addressed.

Additional comments

Thank you for considering my earlier comments. I have no further suggestions. Congratulations on this very interesting work!

·

Basic reporting

The points raised in my last review, on the types of sampling in included studies and on the classification of partial resistance, have been addressed via inclusions in the revised text. Typographical corrections (including a revised URL and a minor code correction) have been applied. I welcome the new text that clarifies the definition of selective sweeps used in this study.

I reiterate my previous opinion: the writing and descriptions are very clear and help the reader to understand the hypotheses and procedures.

Here are some minor typographical corrections:
1. I recommend a search (and replace) for double spaces within sentences (e.g., lines 117, 143, 144, 146 are highlighted by my editing software).
2. Also via search: some hyphenations can be added to compound adjectives (e.g., "antibiotic-resistant" in lines 78, 81).
3. Lines 135, 136: missing colons (or parentheses) for cited studies (see pattern of second sentence).
4. "Furthermore" in line 144 and "Therefore" in line 146 could be deleted without loss of flow.

Experimental design

I welcome the clarified explanation of the bootstrap procedure and I note that the original GitHub code includes empirical p value outputs for the diversity contrasts.

I am also happy with the test of the linear model via simulated data (on GitHub as SimulationsTestGSI_normalized_linearmodel.R and its output GSINormalizedVsUnNormalizedSimData_Res.pdf, and using `lm()` in response to another reviewer's input). Regarding the expectation of higher variance in smaller samples: the point in my first review amounts to the claim that there is a higher chance of finding a spurious association (either positive or negative). This is addressed by the bootstraps, so my point is now moot.

I do note the findings of some <0.05 p values in the linear models of simulated data (albeit with the opposite trend). Therefore it is wise that figure 3 is presented without p values. Although it is referenced in methods, I think it would be wise to reiterate the contrast with simulated data in the results (in lines ~412-414).

Validity of the findings

Owing to the linear model tests, I can now say that my concerns in this section have been addressed, and I consider the findings to be valid.

Additional comments

No comment

---

## Round 0.3 · accepted · Accept

Both reviewers now appear happy with the manuscript, and all comments have been addressed.